# Systemic transphobia and ongoing barriers to healthcare for transgender and nonbinary people: A historical analysis of #TransHealthFail

**Allison J. McLaughlin**[1,2], **Saren Nonoyama**[1¤a], **Lauren Glupe**[1¤b], **Jordon D. Bosse**[1,2¤c]*

**1** Institute for Health Equity and Social Justice, Northeastern University, Boston, Massachusetts, United States of America, **2** School of Nursing, Northeastern University, Boston, Massachusetts, United States of America

¤a Current address: School of Health Professions, Rutgers University, Newark, New Jersey, United States of America
¤b Current address: College of Osteopathic Medicine, University of New England, Biddeford, Maine, United States of America
¤c Current address: College of Nursing, University of Rhode Island, South Kingston, Rhode Island, United States of America

* Jordon.bosse@uri.edu

## Abstract

Transgender (T+) people report negative healthcare experiences such as being misgendered, pathologizing gender, and gatekeeping care, as well as treatment refusal. Less is known about T+ patients' perceptions of interrelated factors associated with, and consequences of, negative experiences. The purpose of this analysis was to explore T+ patients' negative healthcare experiences through Twitter posts using the hashtag #transhealthfail. Publicly available Tweets published between July 2015 and November 2021 from US-based Twitter accounts were collected via Mozdeh. Tweets were deductively analyzed for content using a list of a-priori codes developed from existing literature. Additional codes were developed as new ideas emerged from the data. When possible, type of care location, providers interacted with, and initial reason for seeking care were extracted. Each Tweet was coded by at least two team members using NVivo12. A total of 1,340 tweets from 652 unique Twitter users were analyzed. Negative experiences were reported across healthcare settings and professional types, with physicians, nurses, and counselors/therapists being named most frequently. Primary antecedents of negative healthcare experiences and barriers to accessing care were related to health insurance issues and providers' lack of knowledge, discomfort, and binary gender beliefs. Negative healthcare interactions led T+ patients to perceive receiving a different standard of care and having unmet needs, which could lead to delaying/avoiding care in the future. As such, these results highlight the potential for direct and indirect harm related to providers' specific actions. Patient strategies to prevent and/or manage negative encounters and care facilitators were also identified. A multi-pronged approach addressing healthcare policy, improving knowledge and attitudes of healthcare providers and ancillary staff, and creating clinical settings that are physically and psychologically safe for T+ patients is critical to improving the healthcare experiences, and ultimately health, of T+ people.

**Data availability statement:** The data were obtained via Twitter's (now X) Academic Researcher API and the authors are unable to share it. It is worth noting that several changes have been implemented at X since these data were obtained, including requiring a subscription fee ($100–$5,000/month) for academic researcher access and banning the use of third-party apps (e.g., Mozdeh) to access the data. Interested researchers can apply for access to the X researcher portal at https:// developer.x.com/en/portal/petition/essential/ basic-info

**Funding:** This project was funded in part by an Advancing Health Equity Pilot Grant (JDB) and some writing time was provided through the Humanities Center Fellowship on the theme of Reckoning at Northeastern University (JDB). The sponsors had no role in the study design, collection, analysis, or interpretation of data, development of the publication, or the decision to submit the publication.

**Competing interests:** The authors have declared that no competing interests exist.

## Author summary

Transgender people, who have a gender identity that is different than societal expectations based on their assigned sex at birth, have worse physical and mental health compared to their non-transgender counterparts, which could be due to unequal healthcare treatment. We conducted a historical analysis of Twitter data to healthcare experiences reported by transgender people using #TransHealthFail, which was introduced on Twitter in 2015 to bring attention to the frequently poor treatment experienced by transgender people seeking healthcare. Transgender people reported a range of barriers to healthcare at the level of healthcare systems, including insurance issues and organizational policies. Negative interactions were identified across healthcare setting and provider types, often demonstrating providers' lack of knowledge about, discomfort with, and negative attitudes toward transgender people. Our findings reinforce that negative healthcare experiences have the potential to impact transgender individuals' health both directly and indirectly. Despite general improvements in social attitudes about transgender people since #TransHealthFail was introduced, many of the problems and negative experiences that were identified in the historical tweets are still common themes in more current literature. Our results highlight the critical need to address healthcare issues at all levels of influence to improve the health of transgender people.

## Background

Transgender people, who have a gender identity and/or expression that is different from society's expectations based on their sex assigned at birth – herein abbreviated T+ [1]—have worse physical and mental health compared to their cisgender counterparts (individuals whose gender identity is congruent with their assigned sex at birth). An estimated 0.53% US adults and 1.43% of youth ages 13–17 identify as T+, representing approximately 1.6M US residents [2]. T+ people have higher rates of mental health conditions (e.g., depression and anxiety) [3] and chronic physical conditions [4] like asthma, diabetes, and human immunodeficiency virus (HIV) than cisgender people. T+ people assigned male at birth with a feminine gender identity may have a higher disease burden than other T+ individuals, [5] with Black T+ transfeminine people having the highest risk of mortality compared to both other T+ individuals and cisgender people [6].

One explanation for the disparate health outcomes among T+ individuals is the accumulation of unique stressors due to structural and interpersonal stigma (gender minority stress [7]). Stigma visibility (i.e., visible gender nonconformity) may also increase the risk of experiencing discrimination, [8,9] which, in turn, may increase exposure to risks like violence/victimization. T+ individuals may internalize these negative messages, which can lead to worse mental health directly [10,11] or indirectly through engagement in health-risk behaviors to cope with internal and external stigma [9]. Experiencing marginalization based on multiple minoritized identities may exacerbate the effects of stigma and discrimination on physical and mental health [12]. This is particularly salient as the T+ community is racially and ethnically diverse, with some estimates suggesting a higher proportion of people of color than white identifying as T+ [2,13].

The T+ population faces various issues and barriers accessing healthcare services and navigating the healthcare system. Data obtained in the 2015 US Transgender Survey (USTS) highlighted that one-third of T+ adults reported at least one negative experience with a healthcare provider (HCP) related to their T+ identity in the past year [14]. While high rates

of uninsurance among T+ people may understandably pose financial barriers to healthcare access, even those with insurance frequently experience challenges using their health insurance related to their gender identity and/or expression. The USTS 2015 data also found that, among the nearly 24,000 T+ adults in the US with health insurance coverage, 25% were denied coverage for hormones and 55% were denied coverage for gender affirming surgery [14]. In the current US political climate, T+ youth, and in some states adults, are facing increasing barriers and loss of access to gender affirming treatments (GAT) through anti-transgender policies [15,16]. One way that T+ people (and their allies) have raised awareness of their unique needs and concerns is using social media.

Among T+ people, Twitter (recently rebranded as "X") [17] has been used to share information in real time about positive and negative events and personal contexts like mental health, discrimination, and sexual risk behavior [18]. A common component of tweets is the use of one or more hashtags (#) to serve as an index for the tweet topic(s) and allow the content to be identified more easily. "Hashtag activism", the use of specific # to increase social visibility, has been used by members of minoritized communities to raise awareness of issues unique to their community [19]. For example, #GirlsLikeUs, initiated in 2012, focused on the intersectional experiences of transgender women, has functioned as a way for trans women to build community and to engage with larger cisgender communities about the lived experiences of trans women [19]. Previous research has used publicly available tweets to relate to lesbian, gay, bisexual, transgender, and queer (LGBTQ) healthcare consumers' sentiments about healthcare in association with state-level policy [20]. The use of Twitter data was also demonstrated as a promising source of evaluating consumer reports of healthcare quality [21]. On July 30, 2015, My Trans Health, an organization focused on connecting T+ people with quality healthcare, tweeted: *"We've all experienced a trans health care fail… Share yours with #transhealthfail".* The purpose of this project was to explore publicly available historical Twitter content that includes #transhealthfail, which has been in use for nearly a decade, to categorize tweets about barriers in the process of seeking and obtaining healthcare and the perceived outcomes of negative healthcare interactions on T+ healthcare consumers.

## Materials and methods

An archival search for publicly available Tweets was conducted using the online platform Academic Research Mozdeh (v.5.12.0.0.) [22] in collaboration with a Twitter Developer Premium Account for Academic Researchers. The search was conducted on November 9, 2021 and included original Tweets that a) contained one or more of the following hashtags (#): TransHealthFail, TransDocFail, TransBrokenArm, TransBrokenArmSyndrome (as written and variations all in lowercase letters); b) were written in English, c) posted between July 15, 2015 and November 8, 2021 and d) not duplicate or retweets. Based on available country data in the Mozdeh platform, tweets were further limited to Twitter handles whose users reside in the US. The study was deemed exempt by the IRB at Northeastern University.

**Data Cleaning**. Tweet content was converted to all lowercase letters and references to other Twitter accounts (*e.g., @Username*) were removed. All tweet entry IDs and Twitter handles were converted to unique alphanumeric values. Stop words (a, and, the, etc.) were removed by excluding words that were 3 characters in length or shorter [23]. Finally, the data file containing each tweet's unique ID and cleaned text were imported into NVivo 12 Pro for coding.

**Coding.** A preliminary group of a priori codes were developed from existing literature [24–29] related to individual, interpersonal, communal, and societal factors influencing the health of transgender and nonbinary people; HCP response to gender nonconformity, impact on health and healthcare, and strategies implemented by T+ people to manage their health. Additional codes were added inductively in the coding process. When available in the text of

the tweet, aspects of care such as care location, care need, and role of healthcare professional(s) involved were also coded.

**Data analysis.** Tweets were broken down into smaller groups and each group was coded by at least two team members. In the first read through, tweets were coded for exclusion if they were 1) generally about the #TransHealthFail (*e.g.,* instructing others to search the hashtag and read the Tweets, linking to an article about the trending hashtag) or 2) did not include enough context to understand the experience being described. In all, 269 tweets were excluded, leaving 1,340 tweets in the final analysis. Once each set was coded, the member coding was compared, and discrepancies were discussed until consensus was reached. Codes were grouped and arranged in higher order categories, which were then collapsed and refined into an initial conceptual framework reflecting antecedents, barriers, healthcare experiences, outcomes and consequences of negative healthcare interactions, and patient strategies for navigating the healthcare system.

In the literature review following the analysis, a set of codes developed by Nordmarken and Kelly (2014) about T+ individuals' experiences in the healthcare setting was identified, leading to the refinement of our healthcare experiences category [30]. We reorganized the language within Nordmarken and Kelly's categories to reflect the frequency with which each code appeared (e.g., Invalidation/Misgendering/Exposure became Misgendering/Invalidation/Exposure).

**Positionality.** All members of the interdisciplinary research team have worked directly with patients (medical assistant, student physical therapist, nurse's assistant, and registered nurse) in a variety of clinical settings. Two research team members are cisgender, and one has a family member who is part of the T+ community. Two members of the team are members of the T+ community and have had negative encounters in healthcare settings as both providers and recipients of care. Familiarity with these experiences may have made it easier to recognize more subtle behaviors others may not pick up on. We all also have some higher education, which influences our access to resources such as employment and health insurance. All the team was raised in the Northeast US (and 3 in urban areas), which is generally more progressive than other areas of the country. Collectively, we believe everyone should be able to live as their authentic selves and have access to affordable, high-quality healthcare. We also believe that GAT is both necessary and lifesaving. Throughout the process of coding and data analysis, each team member kept a research journal to document personal thoughts and feelings throughout the process. Individual discussions with the senior author and weekly team meetings were used to discuss questions, concerns, and challenges that arose.

## Results

Most of the tweets used #TransHealthFail and were published in 2015, with about 100 published in the years 2016–2018 and 25 or fewer published each year since 2019. Tweets were posted by 652 unique Twitter handles. Among the accounts in which the specific state could be identified (>90%) all US states except Alabama, Delaware, Mississippi, South Dakota, and Wyoming were represented.

### Care needs, locations, and providers

The most common care needs identified in tweets were mental health services, outpatient, inpatient, and acute emergency services. T+ patients also tweeted about engaging with the healthcare system for diagnostics tests such as labs and imaging, and for the treatment of injuries, pain, various infections, and chronic conditions (e.g., asthma, allergies). Tweets indicated that negative healthcare interactions took place in the context of general healthcare and in the process of

seeking or receiving GATs. T+ people tweeted about their experiences of poor treatment across healthcare settings, including hospitals, urgent care/outpatient clinics, primary care, specialty care, hospital and community-based pharmacies, surgery centers, and waiting areas. The three most common HCP types named in the tweets included physicians, nurses, and mental health professionals (i.e., counselor/therapist, psychologist). Others named as responsible for negative interactions included emergency medical personnel, phlebotomists, pharmacists, specialist services (e.g., anesthesiologists, dentists, endocrinologists, gynecologists, orthopedists, surgeons), ancillary/support staff (e.g., medical assistant), and health professions students.

## Conceptual diagram

Coded tweets were grouped into six categories conceptualizing the experiences described in tweets by T+ healthcare consumers (as represented in **Fig 1**): antecedents, barriers, healthcare experiences, outcomes, consequences, and T+ patient strategies. Within each of the six categories, themes are presented in the order of frequency with which they appeared. Sub-themes that appeared 20 or more times were included as well. Categories, themes, and representative quotes are provided in Table 1.

**Antecedents.** Antecedents were individual and structural factors described by T+ people that were conceptualized to logically precede the barriers and negative interactions that people tweeted about. The most frequently tweeted antecedent was the perception that *HCPs lacked knowledge and/or training* on the specific needs of T+ communities and had *little to*

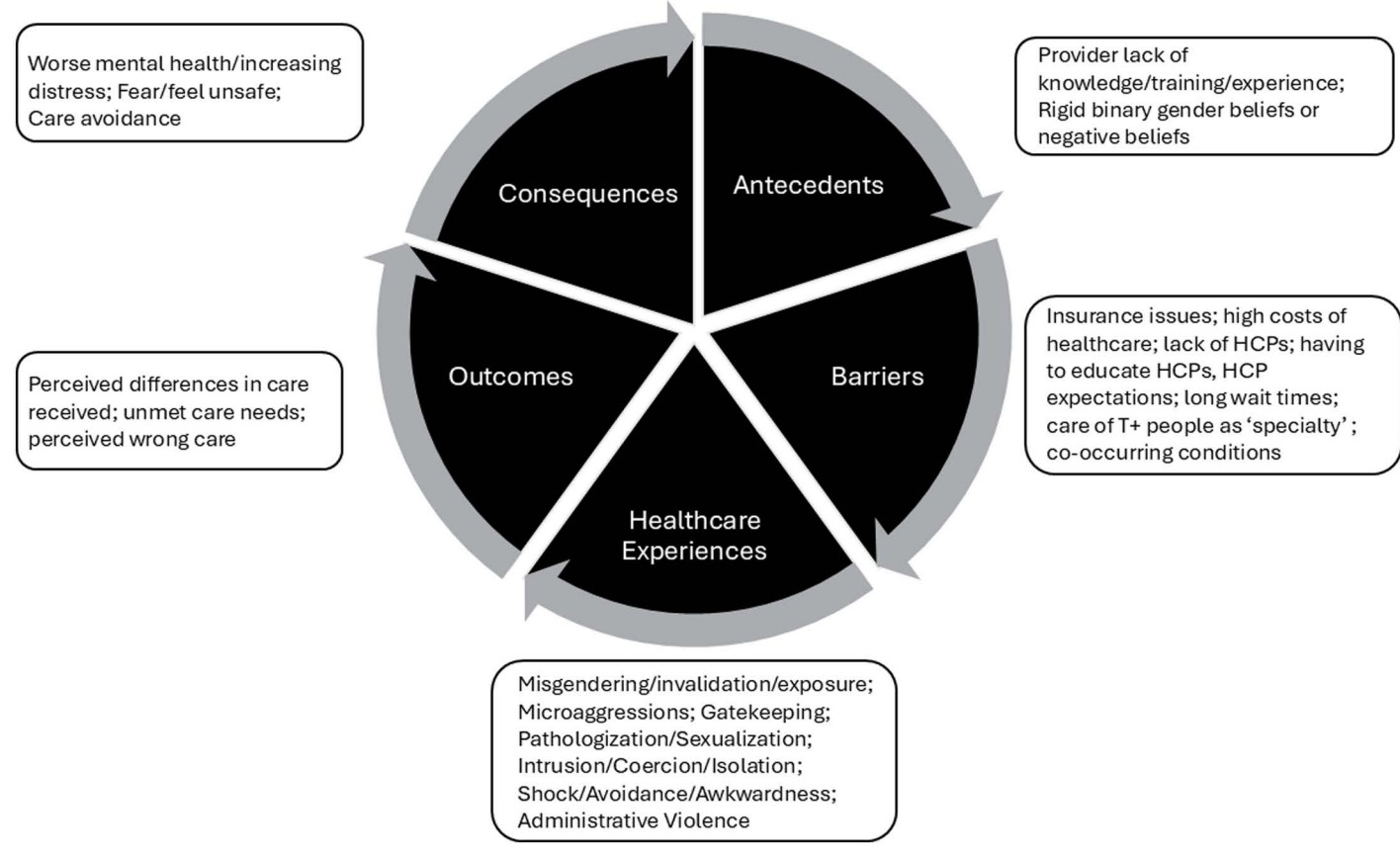

**Fig 1. Conceptual model of T+ people's negative healthcare experiences, outcomes, and consequences.**

**Table 1. Categories, themes, and selected representative quotes.**

| Categories and themes (N=)[A] | Representative quotes[B] |
|---|---|
| **Antecedents (N=186)** | |
| *Provider lack of knowledge/ training/experience* | "No one has any training for trans patients ever" <br> "I once had a nurse who didn't know what transgender or transsexual meant. Like, she didn't even know it was a thing" <br> "We didn't cover trans healthcare in nursing school" <br> "I've never done this. What are your friends' dosages? Do you need bloodwork?" <br> "…ask[ed] me if I was going to start having menses" [Transgender woman] <br> "When I told my psychiatrist I was gender nonconforming, he interpreted me as polyamorous" <br> "Having to explain and educate over and over what transition means to my primary doctor, because 'he's never done this.'" |
| *Rigid binary gender beliefs or negative beliefs* | <u>Rigid binary gender beliefs</u> <br> "…wanted to know why I wasn't dressed fem[inine] enough…" <br> "…women don't wear pants", by a psych[ologist] that was wearing pants herself <br> "…told me I would have to dress in a more "ladylike" manner to access treatment" <br> "…most boys don't want to be skinny" <br> <u>Negative attitudes and beliefs</u> <br> "[HCP] told me if her husband wanted to be a woman, she'd leave him" <br> "Supposedly trans-competent doctor at 6-month follow-up after starting T: 'let's see how this poison is treating your body'" <br> "[Nurse] recommended by LGBT health center tells me that transition care shouldn't be covered by insurance" |
| **Barriers (N=496)** | |
| *Insurance Issues* | "Insurance will finally cover surgery, but not the hair removal required for surgery - which costs hundreds…" <br> "Insurance company refers to transition surgery & HRT as 'medically necessary,' still refuses to grant me coverage" <br> "[Insurance plan] said surgery would be covered, but months later they still haven't provided approval" <br> "Insurance will cover 'breasts' removal but not nipple reconstruction, as it is not 'medically necessary'" <br> "Denying approval for AndroGel because "not FDA-approved as gender affirming hormone" <br> "I may have to fight with my employer because trans health is excluded per my contract" (*categorical exclusions*) <br> "Called [my insurance plan] to ask what trans services are covered, I get [called] "sir" and told 'we don't cover transsexual surgeries'" (*categorical exclusions*) |
| *High costs of healthcare* | "…sucks having my access to hormones I can afford contingent on a blood test I can't [afford]" <br> "I'm out [about] $20,000 out of pocket in spite of being covered" <br> "Choose between feeding family and taking [prescribed] levels of HRT meds" |
| *Lack of HCPs* | "…having to spend hours online or in secret email lists to identify any doctors in your area who will work with you" <br> "…closest provider is 5 hours away" <br> "…no urgent care, doctor, or ER in my city 'feels comfortable' treating me" <br> "[well known surgeon] is already out into 2018" [posted in 2015] |
| *Having to educate HCP* | "Had to explain what transgender was and the treatments for it" <br> "…not getting what I mean when I say transwoman, even after a lot of explaining" <br> "…had to explain to doctors who knew I was trans why I was certain I wasn't pregnant and didn't need a test" <br> "'So you're a lesbian? No? What word do you use? Educate me.' My gender identity was written on the chart in her hands" |
| *HCP expectations* | "Despite hating this terminology, I'm forced to refer to myself as a "male-to-female transsexual" to confused doctors" <br> "I'm just confused about my gender because I haven't known since I was 2 years old" <br> "…said my options were to get 'full male surgery' or 'stay a girl'…" |
| *Long wait times* | "It's been 3y[ears] of useless struggling" <br> "Waited 6 months to see an endocrinologist. Was told 'oh, I don't do trans.' Had to wait 6 more months for another." <br> "Found a superb clinic who said I could start HRT right away. Had to wait another 3 months for my endo[crinologist]'s approval" <br> "used to drive 250 miles for trans care. Finally found a local doc, have to wait 4 months to see her since she is so busy." |
| *Care of T+ people as 'specialty care'* | "…they don't treat 'that issue'…the trans 'issue' that is" <br> "'I don't know how to treat a transgender.' …I'm here about joint pain I've had since before starting HRT?" <br> "Me: 'I'm not feeling well.' Doctor: 'I can't help you transition, [you] need a specialist for that.' Me: 'I transitioned 18 [years] ago.'" <br> "Therapist: 'Sorry but I can't treat transgenders. I can send you to someone else.' Me: 'But I'm here for anxiety from school?'" |
| *Co-occurring conditions* | "…doctor told my insurance my neurodivergence makes me unfit to make my own healthcare decisions" <br> "Doctor says I shouldn't get top surgery because I'm bipolar and 'don't know what I want'" <br> "Depressed and not keeping up on self care? Threatens to call [doctor] and revoke HRT approval" <br> "Had a therapist who refused to give me HRT until they 'cured' my PTSD." <br> "Being denied hormones because the doctor said I 'couldn't know myself' and if I was 'really' trans because I am autistic" |

*(Continued)*

**Table 1.** (Continued)

| Categories and themes (N=)[A] | Representative quotes[B] |
|---|---|
| **Healthcare Experiences** | |
| *Misgendering/Invalidating/ Exposure (N=461)* | Misgendering<br>"…refused to use my name and pronouns"<br>"In the hospital last winter: nurses talking loudly outside room about me being a him, her, it, whatever."<br>"[HCP] misgendering me even if they'd known me only since some time *after* transitioning."<br>"…suggested I might want to attend their men's support group. I'm a trans woman."<br>Invalidation<br>"This is one of my great fears. That a doctor will just declare me 'not trans enough'"<br>"'Would you mind calling me [name]? I'm a trans guy actually.' Gynecologist: 'Well, a guy is not what I am seeing, miss.'"<br>"We only see real women"<br>"My (cis)gender therapist took time out of every appointment to remind me that I wouldn't pass, even with years of HRT'"<br>"'I'm genderqueer and I've had a hysterectomy.' 'Oh so you're a trans man.' 'No.' 'But you had a hysterectomy.' (*misrecognize*)<br>'When I told a therapist I was genderqueer and she responded with, 'So you want to be a man?'" (*misrecognize*)<br>Exposure<br>"Nurse (screaming across a crowded surgery center): 'Well, does he have a uterus?!?'"<br>"Clearly presenting female, forms show birth and current gender. Nurse in crowded waiting room shouts, 'thank you sir!'"<br>"[Being treated] like a sideshow exhibit - literally calling in nurses to come look…"<br>"…in for heart [bypass]. Attending invites all the interns to come 'check out the [slur] and her [gender affirming surgery] results'"<br>"A surgical nurse laughed at me when I panicked about my chest being there, post oral surgery, and outed me to my parents." |
| *Micro-aggressions (N=370)* | "I tell [the doctor] I'm transgender. He asks 'which direction?'"<br>"…sooo do you have a penis or???"<br>"Are you sure you're not a boy?"<br>"Go to ER for hand injury, asked invasive questions about my transition, genitals and 'the surgery' in triage"<br>"What name were you born with?"<br>"When the dermatologist is preoccupied with who and how you can possibly find anyone to date" |
| *Gatekeeping (N=304)* | Deny/Refuse Care<br>"Told me to just leave without even looking at me"<br>"Calling endocrinologists to see if they prescribe trans people HRT and getting called a 'freak' and hung up on"<br>"Having a doctor refuse to treat or see you because of their right-wing religious beliefs"<br>"[HCP]: 'You didn't tell me you were female.' Me: 'I told u I am a transgender man.' [HCP]: 'I can't help you if you lie [to] me. Get out.'"<br>Block/Limit Access<br>"Took 5 years to get on HRT because the gatekeepers decided I was a gay guy, not actually trans"<br>"Doctors refuse to issue the medical reports I need for my official ID change"<br>"Being made to wait three more months to start HRT because the first 13 years 'didn't count'"<br>"Ignoring the continuum of care (2 SRS letters + surgeon okay)…then makes me start the process again. Delay=2.5 years"<br>"Endo[crinologist]: 'I have a gut feeling. 6 more months of therapy'" |
| *Pathologization & Sexualization (N=127)* | Pathologization<br>"Being compared to someone going to multiple doctors for opiate prescriptions for wanting to be on GAT"<br>"…[HCP] didn't know what being trans was. So I told him. He asked if I was having any other hallucinations"<br>"…assuming my manic episodes correspond to my T[estosterone]" (*blame hormones*)<br>"For a heart condition I have had since I was 3: 'Do you think it's the hormones?'" (*blame hormones*)<br>"Eye doctor: 'Probably the hormone's fault you're cross-eyed now.' I've been cross-eyed since birth and on hormones over a year." (*blame hormones*)<br>Sexualization<br>"…told me not to worry so much about surgery because my androgyny is sexy"<br>"…fondled my genitals with no glove and no consent, so basically, I was sexually assaulted by my endocrinologist."<br>"…apparently visual inspection [of breast growth] wasn't enough, so he squeezes them" |
| *Intrusion/Coercion/Isolation (N=111)* | Intrusion<br>"[The doctors] asked if I needed a prostate exam since I'm on T[estosterone]. I came in for a sore throat."<br>"Told my [HCP] I was trans and looking for an endo[crinology] referral. His response: 'Have you considered going to church more often?'"<br>Coercion<br>*While locked in a facility against my will* "Why are you on estrogen? I'm taking you off of it."<br>"ER nurse changed my sex in the records, when I tried to walk out & said it's not safe here, she put me on a psych hold"<br>"Being threatened to stop being treated despite health risks if I argue with the doctor."<br>Isolation<br>"They implied I was a threat, so they put me in a literal corner"<br>"…was put into isolation wing at the emergency room after having a bad bloody nose." |

*(Continued)*

**Table 1.** (Continued)

| Categories and themes (N=)[A] | Representative quotes[B] |
|---|---|
| *Shock/Avoidance/Awkwardness (N=100)* | Shock<br>"The [physician assistant] working for my orthopedic surgeon spent five minutes telling me that she couldn't tell that I 'wasn't a real girl.'"<br>"I told a doctor I was a trans man and for the whole appointment he couldn't grasp that I was FTM not MTF"<br>"Nurse: 'when was your last period?' Me: 'Never. I'm transgender.' Nurse: 'You really fooled me! I thought you were a woman.'"<br>Avoidance<br>"Went to the hospital with an ear infection and the doctor did the exam with tongue depressors so they wouldn't have to touch me"<br>"Had same [doctor] for 3 years when I came out as trans. He told me that he 'didn't know what to do with you people' & left the room"<br>"I went through a 6-hour ER visit without ever being touched by a doctor after I disclosed being transsexual."<br>Awkwardness<br>"The nurse began sweating…and literally stuttered while asking me if I was on T[estosterone] before top surgery"<br>"Nurse was so uncomfortable touching me that she had blood pouring down my arms twice while starting an IV" |
| *Administrative Violence (n=64)* | "Every single medical form I've ever filled out only has binary gender options…"<br>"[Doctor]: 'Get a mammogram and a prostate exam.' Insurance: 'Pick one, based on whether we have an M or an F on your file.'"<br>"…have to wear a wrist band with the wrong name on it after explaining this is traumatic to me"<br>"…have offered name change cert[ificate]…and they ignore it just using my deadname on labels for tests"<br>"That moment when a medical form asking about past menstruation and sexual health excludes trans identities."<br>"Receptionist refused to use my name and pronouns…claiming she needed the legal documents" |
| **Outcomes (N=252)** | |
| *Perceived differences in care received* | "Almost died because they refused to believe it was an allergic reaction and kept focusing on me being trans"<br>"I was hit by a car, and my leg looked like a purple sausage. Was discharged without X-rays and told I 'sprained it'"<br>"…having absolutely no follow-up care after my top surgery"<br>"…Gossiped about other trans patients to me" |
| *Unmet Care needs* | "Was in agonizing…pain and wasn't allowed pain meds"<br>"…didn't treat my depression"<br>"A psych [doctor] I saw to get a letter for HRT avoided talking to me about transition"<br>"…went to the ER to have the doctor recoil at me, barely look at my knee, not ask about my pain, and then misgender me to the nurse" |
| *Perceived wrong care* | "…discharged with instructions to call 'the men's line' for support" [transgender woman] (*wrong referral*)<br>"…changed the gender marker on the forms to misgender me" (*wrong documentation*)<br>"…those are just stretch marks" [when I had cellulitis] (*wrong diagnosis*)<br>" …gallstones misdiagnosed for years because ER doc said my abdominal pain was due to binding…"<br>"Being told it makes no difference if I take my estrogen sublingually or not" (*wrong education*)<br>"Seeking help for suicidal ideation, misgendered by staff, placed in wrong gender room, even though gender legally corrected" (*wrong room*) |
| **Consequences (N=239)** | |
| *Worse mental health/ increasing distress* | "…made me feel more suicidal as a whole and blamed me for everything happening in my life…"<br>"…broke down crying during the exam"<br>"Having to take an entire day off for an OB/GYN exam to deal with the emotional exhaustion and stress"<br>"Hint...putting a suicidal trans woman in an environment that aggravates their dysphoria negates the efficacy of the treatment" |
| *Fear/feel unsafe* | "…too scared of how I'd be treated in the ER to go"<br>"…terrified of what may happen if I were unconscious"<br>"Choosing between health and the real possibility of being mistreated by transphobes"<br>"I've never felt safe or comfortable enough to come out to a doctor"<br>"Being afraid to be honest with your therapist in fear that they won't give you an HRT letter"<br>"…afraid to ask my dentist for jaw/facial feminization referrals for fear of being dropped" |
| *Care avoidance* | "Not asking for medical help with sensitive issues involving your genitalia [because] doctors will humiliate you"<br>"…don't seek care because of negative experiences and discrimination"<br>"Chose not to go to psych ward"<br>"Not going to the doctor for 10 years except emergencies because I don't want to deal with the bull shit"<br>"Avoided going to the doctor for bothersome yet easily treatable problems, because it's easier than explaining 'trans'"<br>"Letting other health problems go undiagnosed and untreated because transition is more important" (*prioritizing GAT*) |
| **T+ Patient Strategies (N=255)** | |
| *Deciding when to disclose gender identity/history* | "Years ago, I stopped telling doctors my history, unless it was relevant. Not ideal, neither is being refused treatment."<br>"Never telling a medical professional I'm on HRT as they blame that immediately"<br>"Not telling any medical professional that I'm trans because their attitude instantly changes for the worst when I do"<br>"Outed my partner (w/ permission) to a doctor as trans. Gave correct pronouns" |

*(Continued)*

**Table 1.** (Continued)

| Categories and themes (N=)[A] | Representative quotes[B] |
|---|---|
| *Conforming to provider expectations* | "Lying about your 'always knew' narrative so you're more likely to get approved for surgery"<br>"Must live as your psychiatrist's stereotypical ideas about your gender role for 1 year before being approved"<br>"Giving prescribers the answers you think they want to hear so you'll get the care you need" |
| *Advocate for themselves* | "My partner corrected her on my name"<br>"Now I'm seeing another doctor"<br>"My primary when I came out barely even knew what trans was. Changed primaries out of concern at lack of familiarity" |
| *Educate themselves and others* | "Presenting my doctors…with peer reviewed HRT info I found…that they had never seen"<br>"Go to nurse for help w 1st T[estosterone] injection, tells me she doesn't know how. I go home and watch YouTube videos, do it myself" |
| *Finding other sources of treatment* | "…prefer mail-order for their pharmacy orders"<br>"Started self-medicating because I couldn't find a therapist … best choice I made."<br>"Ordering hormones online from other countries and hoping they're not counterfeit"<br>"Ordering progesterone overseas because the pharmacy hiked the price from $50 to $75 and insurance won't cover any of it" |

**Notes**

[A]N's for each category reflect all responses in the category, but only themes and sub-themes with >20 responses are reflected in the table. To minimize confusion, we do not include the number of responses per theme/sub-theme in most categories except for healthcare experiences.

[B]We have retained the original language/abbreviations used in the Tweets but would like to point out that these are not always the most current language. For example, "hormone replacement therapy" and "sex reassignment surgery" are more commonly referred to as gender affirming hormones and surgery (or treatments, more broadly). Similarly, acronyms such as FTM and MTF are less frequently used today but may still be the language of choice for some older T+ individuals.

Abbreviations: ER, emergency room; FDA, food and drug administration; FTM, female-to-male; GAT, gender affirming treatment; HCP, healthcare provider/professional; HIV, human immunodeficiency virus; HRT, hormone replacement therapy; LGBT, lesbian, gay, bisexual, transgender; MTF, male-to-female; PTSD, post-traumatic stress disorder; SRS, sex reassignment surgery.

*no prior experience* providing direct care to T+ people. Tweets also indicated that T+ people experienced HCPs as holding *rigid, binary gender beliefs*, such as wanting patients to choose a binary gender or demonstrate a gender expression consistent with stereotypical standards or expectations of their affirmed gender (e.g., wearing make-up). Some tweeted about generally *negative attitudes and beliefs* regarding T+ people that they heard expressed during healthcare interactions (e.g., HCP referring to hormones as 'poison').

**Barriers.** Barriers were conceptualized as issues T+ people described that prevented them from either accessing healthcare or receiving an appropriate standard of care; *insurance issues* and *cost of healthcare* were the most tweeted about. Many T+ people highlighted a lack of health insurance coverage, which included *categorical exclusions of GATs*, often deeming them "cosmetic" or "elective." Other significant barriers described were a *lack of HCPs* and *long wait times.* T+ care recipients also tweeted about *having to educate HCPs* on what it means to be transgender and their treatment needs as a barrier to care. Another barrier was the perceived *HCP expectation* about what it means to be T+, which included expecting a monolithic story (e.g., being 'trapped' in the wrong body and/or desire for genital surgery) and stereotyped gender conformity (e.g., hyperfemininity for a trans woman). Care for T+ people, even when unrelated to gender, was often seen as *specialty care* under the purview of a different HCP. Finally, co-occurring, often pre-existing, physical and mental health conditions also complicated the pursuit of GAT. In many of the tweets, T+ people perceived that having an existing mental health condition such as depression, bipolar disorder, eating disorders, autism spectrum disorders, post-traumatic stress disorder, substance use disorders, or suicidality, was a barrier to their accessing GAT.

**Healthcare experiences.** Within the broader category of healthcare experiences, tweet content was organized into one of seven themes, which are presented in frequency order: misgendering/invalidation/exposure, microaggressions, gatekeeping, pathologization/sexualization, intrusion/coercion/isolation, administrative violence, and shock/awkwardness/avoidance.

Misgendering/Invalidation/Exposure:    *Misgendering, invalidation, and exposure* were the most common themes of healthcare experience. Most frequently, people tweeted about being *misgendered*, referred to by incorrect pronouns or form of address based on gender identity/ expression. T+ people also described *invalidating* statements such as being told they're not "really" trans or "not trans enough", not a "real" man or woman, or are in a "phase". *Invalidation* included when HCPs *misrecognized* individuals (often those who identified as gender-queer) and either "assumed [a T+ patient] to embody a normative binary identity, or could not recognize how to categorize them"(p.354).[27] At times, T+ people perceived that HCPs had difficulty matching up the T+ patient's presentation with the gender marker in their chart and that HCPs became confused when a patient was seeking sex/gender specific care and the HCP had difficulty understanding the patient's anatomy (e.g., a T+ man presenting for a gynecological exam). T+ people described experiences of *exposure* when their personal health information, anatomy, gender identity and/or gender history were revealed by HCPs to others who did not need to know, often in public settings such as in the waiting room or an open emergency room. T+ patients also tweeted that HCPs brought in coworkers, students, or other people unnecessarily (and often without permission) for observation, which was another form of exposure.

Microaggressions:    *Microaggressions* were defined as 'interactions that communicate "othering" messages, referencing gender nonconformity, that makes transness an issue, or that can cause [T+] people to feel self-conscious of their transness"(p.145) [30]. *Unnecessary or inappropriate behaviors, statements, and questions* were the most common *microaggressions* that were tweeted about. For example, T+ people described HCPs offering or requiring irrelevant tests or exams and asking unnecessary questions about assigned sex at birth or other gender-related information when they had a medical need unrelated to their gender (e.g., sore throat, cold, broken bone). T+ patients also tweeted that they were referred to as "transgenders" (as a noun), other slurs, or outdated language.

Gatekeeping:    *Gatekeeping* encompasses tweets describing HCPs blocking or limiting access to GAT on the basis that the individual was "not ready" or did not fully meet strict eligibility criteria [31]. The most frequent sub-theme was *care refusal.* Examples included the outright refusal to serve T+ patients by any HCP type (e.g., physician, pharmacist, etc.) or a perceived unwillingness to provide GA care specifically. Tweets suggested that HCPs *blocked or limited access* to GAT at times by making patients wait substantial amounts of time to be prescribed hormones or pass evaluations. Insurance companies were also perceived as responsible for *gatekeeping* by refusing to cover requested treatments or creating stringent requirements for GAT to be deemed medically necessary.

Pathologize/Sexualization:    *Pathologizing behaviors* treated one's transness as medically abnormal or unhealthy or defined one's gender identity, expression, or hormones as the *blame* for other health issues. Most commonly, T+ patients reported their hormones were blamed for symptoms that they reported were present prior to initiation of hormones. Some tweets suggested the recommended treatment would be to decrease or stop hormone doses, regardless of any association between hormones and symptoms or if symptoms were known side effects of hormones (e.g., acne with testosterone). T+ individuals also indicated their attempts to secure GAT were pathologized, for example, being likened to "doctor shopping" by someone dependent on opioids. *Sexualization* was used when T+ people described HCPs' questions and comments to be focused on T+ patients' attractiveness, genitals, and/or sexual activity that were seemingly unrelated to the T+ patient's presenting problem.

Intrusion/Coercion/Isolation:    *Intrusion* involved T+ patients perceiving an HCP as disciplining or chastising them or utilizing religion against them. Tweets included HCPs telling

those requesting GAT that they should pray, go to church, or read the Bible. T+ patients felt chastised for making HCPs uncomfortable when correcting behaviors such as deadnaming. *Coercion* occurred when T+ patients felt HCPs did not provide choices regarding their care, including insisting that they receive unnecessary STD testing as a requirement of care. *Coercion* also included when T+ patients perceived they were provided limited options for treatment, particularly all-or-nothing approaches (e.g., requiring an interest in pursuing all forms of gender affirming surgery, including bottom surgery). In the most extreme tweets regarding *coercion*, T+ care seekers described unnecessary genital exams/sexual assault, abuse, threats, and involuntary medication/restraint. T+ people also tweeted about *isolation* in emergency and mental healthcare settings and described being physically kept apart from other patients without reasonable explanation.

Shock/Avoidance/Awkwardness:    *Shock, avoidance, and awkwardness* were used to categorize T+ patients' tweets about HCPs reacting in ways that portrayed discomfort. T+ patients perceived HCPs responses such as *shock* and becoming *awkward*, demonstrating a change in body language and attitude upon learning that the patient was T+. Some T+ patients reported that HCPs employed *avoidance* when uncomfortable and/or disengaged by walking out of the room, recoiling, or refusing to directly touch the patient.

Administrative violence:    *Administrative violence* [32] was used to describe ways in which the classification of gender was experienced as harmful to T+ people seeking healthcare services (e.g., environmental or systemic microaggressions). One of the main forms of administrative violence was the use of only binary gender options on forms or in the electronic health record (EHR). Related, questions specific to the wrong sex (either assigned or affirmed) during assessments that were not applicable made T+ people feel hyperaware of their differences. For example, trans women received automated EHR reminders for cervical screenings, questions about their last menstrual cycle, and had to explain that they do not menstruate. Some tweets indicated that health systems' refusal to update names or gender markers in the EHR, even with legal documentation, contributed to HCP confusion.

## Outcomes

Outcomes referred to T+ people's experiences that were conceptualized to have occurred as a result of negative healthcare interactions. A common outcome among tweets was *perceived differences in care received,* where T+ patients posited receiving care that was inferior to and/or inconsistent with an expected standard of care. Tweets described being discharged from the ER without what felt like proper evaluation and being discharged from surgery without follow-up. *Perceived differences in care received* also included tweets that suggested some HCPs may lack professionalism (e.g., HCPs disclosing information about other T+ patients without their consent).

Another common outcome was *unmet care needs,* when T+ patients perceived their health needs were not addressed. For example, T+ patients reported being denied pain medication for severe injuries. Some T+ individuals shared concerns about mental health providers who they felt did not adequately treat their symptoms or fulfill their gender care needs. For example, HCPs refusing to talk about trans-related issues when that was the patient's sole reason for seeking mental health care or refusing to provide a letter of recommendation for GAT.

Many T+ patients thought they *received the wrong care*. This included what were seen as wrong referrals, inaccurate diagnoses, incorrect information, wrong rooms (i.e., being assigned to a room based on assigned sex at birth versus affirmed sex/gender), or incorrect documentation (e.g., wrong name on EHR, prescription labels, and hospital ID bracelets, even after legal name changes).

## Consequences

Consequences were defined as long-term impacts on T+ people's health, wellbeing, and help-seeking behaviors. The most frequently mentioned consequence was *worse mental health* or *increasing distress,* which included descriptions of worsening gender dysphoria and feeling emotionally exhausted, overwhelmed, or panicked. Some T+ patients with pre-existing mental health conditions also described increasing feelings of depression or suicidality specifically in relation to their experiences of HCPs' behaviors such as deadnaming, misgendering, and care refusal. Reports of stress or anxiety around gender-specific care were common, especially with invasive gynecological examinations.

Additionally, T+ patients tweeted about *fear* surrounding the uncertainty of HCPs' reactions to their T+ identity and the potential subsequent (mis)treatment. T+ patients seeking referrals for GAT shared *fears* of being deemed 'mentally ill' or being 'dropped' by their HCP. *Fear* of losing access to GATs, specifically, was also common among tweets, leading some patients to minimize or deny symptoms to maintain access to the GAT they did receive. T+ people also tweeted about *care avoidance,* delaying or avoiding necessary medical treatments out of fear of discrimination. Some T+ people described avoiding mental healthcare despite suicidal thoughts, due to fear of losing access to hormones or being placed in a room that did not align with their gender identity.

## T+ patient strategies

T+ patients also tweeted about *strategies* used to prevent and/or navigate healthcare interactions and systems considering the antecedents, barriers, healthcare experiences, outcomes, and consequences that were identified. A predominant strategy highlighted in the tweets was *deciding when to disclose their gender identity and/or gender history* and to whom. Another strategy was to keep copies of legal documents such as name or gender marker changes on hand. T+ patients also described *conforming to provider expectations*, including telling the 'story' of their gender identity (and presence of gender dysphoria), using specific language to describe themselves, and/or presenting their gender in a binary way. T+ patients shared stories of 'jumping through the hoops', even those that they did not agree with, to demonstrate they met criteria for GAT. T+ patients also *advocated for themselves* by correcting HCPs when they used the wrong name or pronouns. When HCPs were perceived as lacking knowledge of care needs, T+ patients *educated themselves* and frequently *educated the HCP* as well. Some T+ patients found *other sources of treatment* (e.g., mail order pharmacies; HCPs in other countries) to work around limited local systems or to 'self-medicate.'

## Discussion

Our analysis sought to categorize T+ people's tweets about their negative healthcare experiences using #TransHealthFail. We further conceptualized potential antecedents, barriers, outcomes and consequences of these negative interactions, as well as T+ patients' reported strategies for navigating healthcare across these categories. While the systemic and structural nature of transphobia has been identified as a significant factor in T+ individuals' healthcare experiences previously, [24,33] these findings provide explicit examples for the various ways T+ people experience stigma at structural, interpersonal, and individual levels. T+ individuals described negative experiences seeking and receiving both general healthcare and GAT, across treatment settings and HCP types. Worth noting, when T+ patients were receiving care that was billed as LGBTQ+ healthcare (or T+ healthcare more specifically) or in environments that demonstrated some awareness of T+ people (e.g., inclusive forms), there was both a certain

expectation of competence in care provision to T+ people and greater disappointment when negative experiences occurred.

The conceptual model (Fig 1) that was developed highlights the cyclical and potentially non-linear relationships between the categories of described experience and the cross-cutting nature of T+ patient strategies. Worth noting, the categories and their relationships are conceptualized based on the corpus of tweets and what could be interpreted from overlaps in various data. Some categories may be related to other categories directly or indirectly. Perceptions categorized as antecedents (*provider lack of knowledge/training/education* and *negative beliefs*), for example, could directly influence barriers (i.e., *lack of trained providers*), healthcare experiences (e.g., *microaggressions*, *care refusal*), outcomes (e.g., *unmet care needs*), consequences (e.g., *care avoidance*) and could lead to the development of T+ patient strategies (e.g., *advocating for themselves, educating themselves and others, finding other sources of treatment*). Meanwhile, some people may not have had relevant antecedents or barriers, but still had negative HC interactions. Some T+ patients may have avoided or delayed care based solely on hearing about others' experiences, not their own direct experiences [34]. Not all individuals who tweeted about negative HC experiences may have experienced the outcomes or consequences that were described, and consequences (i.e., *fear, care avoidance*) could have been due to other factors such as experiencing discrimination in non-healthcare environments [35]. As such, the following discussion of the study findings are presented with these nuances in mind.

**Antecedents.** HCPs' perceived lack of knowledge, negative attitudes, and outdated or stereotypical gender beliefs, were experienced by T+ people as feeling invisible or othered during healthcare interactions. Existing research suggests that HCPs' lack of knowledge or outdated ideas could be related to lack of formal training in the curricula of most health professions' education, [36,37] directly contributing to uncomfortable clinical experiences for both patients and providers [38]. Prior research suggests that some HCPs do not see the importance of knowing someone's gender identity, asserting that they 'treat everyone the same', yet they may still enact microaggressions toward T+ patients [39]. Indeed, people who think they 'don't see gender' (or skin color or other stigmatized characteristics) not only ignore salient identity traits, [40] but may act in a manner that is *more* biased than those that acknowledge differences between individuals [41].

Many tweets highlighted the impact of HCPs' perceived outdated ideas on transness, resulting in expectations of T+ patients to tell a specific, monolithic narrative of distress [42] misery at being 'trapped in the wrong body', [43] and to 'perform narrative binary gender' [44] for access to GAT. In addition, some HCPs were perceived as expecting a gender normative expression to demonstrate appropriate 'fit' for GATs ([45] Worth noting, the majority of the tweets in which T+ patients described being encouraged (or required) to conform to stereotyped gender expression, were made by transgender women. Transmisogyny, the intersection of misogyny and transphobia, [46] is a specific barrier to care for trans women [47]. Transmisogyny may also manifest in behaviors such as sexualization of transgender women, [47] which was described by some T+ individuals in this analysis.

An emphasis on binary gender identity and presentation, as described in many tweets, reinforces cisnormativity and affords more social power to T+ people who can blend (i.e., "pass") as a member of a specific, binary gender, [48] which is not a priority or desire for every T+ person. Adherence to a rigid gender binary may also influence care experiences for T+ patients who are nonbinary. For example, existing evidence suggests some HCPs feeling uncomfortable prescribing gender affirming hormones to individuals with a gender identity outside the gender binary, [49] in favor of a recognizable 'transition' in which one "goes all the way" [50]. Being recognizable as T+ (i.e., stigma visibility) may also increase the likelihood that individuals will experience mistreatment in healthcare settings, particularly among

services that are historically gender-based (e.g., rape crisis and domestic violence treatment) and mental health settings [51].

**Barriers.** A significant barrier at the systemic level, perceived by T+ individuals who had health insurance, was the lack of coverage of GAT services. Though there has been some progress in this area, T+ people continue to be un- or under-insured at higher rates than their cisgender counterparts [52]. Those who are insured may still face insurance-related barriers [14]. For example, a 2022 analysis of over 200 health insurance plans in 33 states found only half of the policies explicitly *included* coverage for gender dysphoria [53]. However, policies that cover GATs may not cover the full extent of procedures outlined in treatment guidelines [54]. In a 2022 survey of over 92,000 US transgender adults, one in four respondents reported having insurance issues related to GATs in the past year [55]. Requiring a formal diagnosis of gender dysphoria (and gender identity disorder previously) has proven to be a double-edged sword for T+ people. Though a diagnosis legitimizes one's request for, and can facilitate access to GAT in a US-based medical system that relies on diagnosis codes, it has been reported that the presence of the diagnosis has been used by insurance companies to deny coverage for non-gender related care; sometimes known as "trans broken arm syndrome" [56].

Though, historically, GATs were initiated in specialized gender clinics (often associated with academic hospitals), they can be provided in the primary care setting. Situating GAT outside the purview of primary care and general medical practice allows HCPs working in such settings to remain ignorant about T+ care needs (albeit T+ people also need routine medical care) and leaves fewer provider options for those seeking GAT. Fewer options for treatment make it more difficult to obtain care by creating longer wait times and requiring long-distance travel. Limited treatment options can also put a heavier burden on HCPs who are willing to provide GAT, leading some HCPs to provide GAT secretly, so as not to be overwhelmed with the high demand for treatment [45]. Finally, categorizing GAT as 'specialty care' may suggest to some HCPs that they do not *already* know how to care for T+ patients. For example, one tweet described a nurse not knowing how to administer gender affirming hormones (e.g., injectable testosterone or estradiol) because they have no prior experience. However, these hormones are delivered in the same manner as other injectable medications and administering injections by any route is an entry-level nursing skill [57].

Further, co-occurring conditions, both physical and mental, were characterized as barriers to care. Though co-occurring mental health conditions were perceived to be a common reason for HCPs to delay or refuse care, recent research has demonstrated that access to GAT such as puberty blockers, hormones, and surgery can lead to improved mental health by decreasing depression and anxiety, reducing body uneasiness, and improving life satisfaction, body satisfaction, and quality of life [58,59]. Physical changes from hormones and surgical interventions may improve mental health by increasing the amount that others perceive the T+ individual as belonging to their identified gender (visual confirmation of affirmed gender) and decreasing the amount of body-related dysphoria experienced [60].

**Healthcare Experiences.** Negative interactions were reported across clinical settings and HCP types, highlighting the ubiquity of systemic and interpersonal stigma. The most common negative interactions T+ people tweeted about involved misgendering, invalidation, and exposure. Misgendering is a common experience for T+ people in many healthcare settings, [61,62] even in those that provide GAT [63]. The experience of being misgendered can be detrimental to T+ people's mental health [62]. Misrecognition, which has previously been described as making incorrect assumptions about T+ individuals' bodies or identities, [27] is a potentially more subtle form of invalidation. This experience was reported frequently among patients with genderqueer identities, when HCPs were perceived to both have some understanding of the T+ patient's transness and attempt to put them in a binary gender box (e.g., assuming a

genderqueer person assigned female at birth who had a hysterectomy is a transgender man). These findings support existing research that suggest genderqueer or nonbinary individuals more commonly face misrecognition in healthcare settings [64]. It is also worth noting that invalidating experiences of misrecognition that were described are rooted in cissexism and heteronormativity and can affect healthcare interactions for cisgender partners of T+ people as well [65,66]. For example, a cisgender woman with a transmasculine partner may be pressured to take birth control or undergo a pregnancy test, because the HCP does not believe that it is possible for her to be partnered with a man and not have the potential to be pregnant.

T+ patients described being put on display, having unwanted attention brought to their gender identity or gender history. Some tweets indicated that HCPs publicly disclosed T+ patients' gender identity or gender history in public areas like waiting rooms and ridiculed T+ patients by gossiping, laughing, and mocking among colleagues. These examples of exploitation and exposure violate patients' rights to privacy and being treated with dignity and respect. Additionally, T+ patients perceived being implicitly expected to use their bodies to educate others (e.g., medical and nursing students) and serve as a form of objectification [45,67]. Such privacy violations can both increase T+ patients' vulnerability [68] and have the potential to cause direct and indirect harm to T+ patients (e.g., worse mental health, care avoidance, fear).

Further, many T+ folks seeking care tweeted that they were refused care, which is a well-documented occurrence for T+ people seeking healthcare [14,25,69,70]. Among T+ people who were not refused care, negative healthcare interactions remain common despite nearly a decade having passed since #TransHealthFail was introduced. Most recently, among 1,684 T+ adults assigned intersex or female at birth who had engaged in healthcare in the prior year, 70% reported having some kind of negative experience, including having to educate their HCP about gender identity to receive care, being asked inappropriate questions, being refused both general medical and GAT, harsh verbal language from the HCP, being misgendered and/or deadnamed [71]. In nearly all categories of negative interactions, people who had sought GAT reported these experiences more frequently compared to those who had not sought GAT [71]. As such, the pursuit of GATs, specifically, may be a form of stigma visibility with the potential to increase some T+ individuals' risk of being treated poorly in healthcare settings.

Frequently, T+ people, their gender identity, and/or their use of GAT such as hormones were seen as the problem, often being attributed as the cause for co-occurring health issues [56]. T+ patients perceived the result of such pathologization was the HCP wanting to lower hormone dosages or stop hormone use entirely. One proposed role of HCP behaviors such as blaming, shaming, and othering is to re-establish the power of the HCP in a healthcare encounter, [72] and some tweets portrayed the HCP's ability to stop prescribing hormones as a means of holding power over T+ patients. A small number of tweets also referred to experiences of being sexualized by the provider. Other research has highlighted T+ patients experiences like frequent physical examinations and questions about sexual activity that were not related to the visit [29]. These findings are also supported by a large, national study of transgender and nonbinary adults, in which 1.2% (n=245) reported they had experienced unwanted sexual contact in the healthcare setting [73].

Finally, the data suggested that systemic barriers (i.e., administrative violence) contributed to the negative healthcare experiences of T+ people. The inability to be recognized by health systems due to lack of inclusion serves to erase the possibility of T+ people existing in the system altogether [30,74]. Having their identities and experiences excluded in cisgender-centric forms, EHRs, assessment questions, and/or rooming policies was distressing for T+ patients. These structural issues could lead to negative healthcare experiences even when individual HCPs within the system aim to treat T+ patients in an affirming manner. As such, systemic solutions to facilitate positive healthcare encounters for T+ people are necessary.

**Outcomes.** Primary outcomes from T+ individuals' negative interactions included the perception that they received a different standard of care, the wrong kind of care, and/or had unmet care needs. These findings are consistent with prior research that has demonstrated that LGBTQ+ people across the life span may receive lower quality care than their cisgender, heterosexual counterparts [75–78]. T+ patients reported not having care needs met for general medical concerns (i.e., acute injury/pain, infection), as well as requests for GAT, including obtaining letters of recommendation. Unlike prior research, which has identified cost of care as the main contributor to unmet care needs, [79] these findings highlight the potential role of HCP-specific barriers in unmet care needs. T+ patients thought HCPs' negative attitudes and lack of knowledge also contributed to what they perceived as missed diagnoses and receiving a lower standard of treatment, while institutional policies contributed to errors such as being placed in the wrong room (based on assigned sex rather than gender). Combined, these results suggest the need for a multi-pronged approach to improving the healthcare experience of T+ people across healthcare settings.

**Consequences.** One of the most common consequences of HCPs' perceived negative behaviors included worsening of distress or mental health. Negative HC interactions (and hearing about others') can lead to the development of distrust and lack of feeling safe in the HC environments [28]. Tweets suggested that T+ folk developed negative expectations about future HC interactions, often resulting in avoidance or delay of care, which is consistent with the tenets of both Minority Stress Theory [7,80] and prior studies of T+ individuals' healthcare experiences [81,82]. For example, about a quarter (23%) of respondents in one sample reported avoiding healthcare due to fear of discrimination [14]. Another analysis of the same large data set highlighted the direct relationship between negative interactions like invasive questions, care refusal, and having to educate their HCP and increased odds of reporting healthcare delay [81]. To some extent, the chronic, ongoing nature of negative healthcare interactions may contribute to feelings of defeat or hopelessness that the system will ever improve, which could also contribute to worse mental health. For example, in one study, T+ individuals who had avoided healthcare specifically due to fear of discrimination reported worse mental health than those who avoided healthcare for any other reason [83]. Finally, another way that fear manifested was through T+ people internalizing an expectation to tolerate unacceptable HCP behaviors so as not to be deemed "difficult" and/or risk losing access to care.

**Strategies.** The main strategy for self-protection identified in the data was identity management, choosing whether (and when) to disclose one's gender identity or gender history, which has been demonstrated in prior research [84,85]. In addition to non-disclosure of identity, not speaking up about specific needs and tolerating interactions to get the care they needed were also employed as ways of protecting limited access to care [25,72]. Alternately, T+ individuals structured their narratives to meet what they perceived as the providers' expectations so as to not raise any 'red flags'. Previous research suggests this strategy has been employed specifically to get needs met, [86] and may be particularly relevant to nonbinary individuals [87]. T+ care seekers also tweeted about setting boundaries (e.g., correcting providers, changing HCPs when possible) and advocated for themselves during interactions and more publicly, demonstrating T+ care-seekers' resilience [88].

Worth noting about resilience strategies is that several behaviors considered strategies for some (e.g., educating HCPs or traveling far to see an affirming HCP) were seen as barriers to care for others. To help minimize the occurrence of negative interactions and navigate the complex care environment, T+ care seekers reported they relied on formal and informal community networks to share information about HCPs, the process for seeking care, and workarounds to help overcome barriers, which has been described previously [70]. One way

to draw on both the strengths of the T+ community and the benefits of social and tangible support (e.g., assistance getting to an appointment) is to collaborate with community health services and providers to implement programs (e.g., buddy systems) so T+ individuals do not have to attend visits alone [89].

**Limitations.** A significant limitation of this qualitative analysis is that the data, by nature of them being retrieved from Twitter, are in a condensed format (140 characters and spaces), which is likely not able to characterize people's full experiences. However, this approach did allow us to obtain a large body of tweets from individuals posting in response to the hashtag. Tweets are T+ patients' descriptions and perceptions and are not a direct observation of those healthcare experiences. As such, the exact behavior, knowledge, attitudes, or beliefs of HCPs is not known. At the same time, it is important to acknowledge that people's perceptions are their reality, and perceived bias can have a significant impact on health and health-seeking behaviors [90]. Also unknown to us from these data are the range of identities and lived experiences that could be relevant to the T+ people tweeting. Though the bulk of the tweets are specific to being T+, some of the barriers (e.g., insurance-related issues, high costs of health care) and experiences like microaggression are systemic in nature and occur more commonly among people with marginalized identities. Similarly, the outcomes T+ people described (e.g., perceived lower quality care and unmet care needs) could also be related to other intersecting identities. As such, it is also important to further explore the intersections of other identities lived experiences (e.g., race/ethnicity, under-insured, unstably housed, etc.] and negative healthcare experiences.

The data obtained for this analysis only included tweets with a relevant hashtag. People may have responded without including the hashtag whose tweets would not have been identified. These data are only from public accounts, and we do not know how many more tweets were published in private accounts and whether they reflect similar or different themes in experiences. Demographic data were not available, so whether or to what extent the individuals whose tweets were analyzed represent the general population of the US is also unknown. In 2015, when the bulk of the analyzed tweets were disseminated, 72% of US adult internet users used Facebook and only 23% used Twitter, with a higher percentage of Twitter users being Black or Hispanic [91]. Examining content related to T+ health on other social media platforms that also use hashtags and have a different user demographic (e.g., Facebook, Instagram) could be beneficial to identify needs and experiences by different subgroups. Particular attention should be paid to multiply marginalized T+ individuals (e.g., racialized groups, individuals with disabilities, etc.) who may experience more discrimination [69] or healthcare denials, [92] have access to fewer resources, [93,94] and face higher risk of mortality [6]. Similarly, because identity data are not available, it is difficult to discern whether any of the tweets were from individuals with nonbinary, agender, and genderqueer identities or examine potential differences in needs and experiences. Some evidence suggests that nonbinary respondents reported lower uptake or future interest in GATs as well as lower healthcare utilization for both annual wellness visits and mental health services [95]. This could be an important area for future research.

Data were obtained from Twitter handles who listed their location as the US or had a location listed that was identifiable as the US. This was done intentionally because the healthcare system in the US, unlike most other countries, is privatized. Though it is possible that T+ people in other areas of the world have similar healthcare experiences as well as unique differences, we cannot be sure from these data. A comparison of data obtained from multiple countries could identify experiences that may be more universal. Similarly, there were not enough data from each state to allow us to explore differences either within or between states. It is possible that T+ individuals' HC experiences will vary based on state-level policies

guiding the treatment of T+ people and access to GAT. In addition, T+ people living in rural areas face unique challenges such as increased visibility in tight-knit communities, fewer healthcare services and support structures, and conservative anti-transgender laws [96]. Further research is needed to explore the rural/urban differences in healthcare experiences among the T+ population..

Another potential limitation of Twitter analysis came from the occasional overlap of tweets describing similar ideas and fitting into multiple categories. While this helped address the multifaceted nature of individual experiences, it made it difficult to keep track of the exact frequency of different categories and made some of the similar sub-categories more difficult to distinguish. However, the overall order of the seven themes within healthcare experiences remained clear due to large enough differences in their frequency. Finally, developing a conceptual model of HC experience was challenging because, as noted previously, experiences were not necessarily linear. It is also possible that occurrences we deemed as outcomes were precursors to negative interactions. For example, one study of T+ adults identified that individuals with psychological distress or suicidal ideation were less likely to have been treated with respect by an HCP [97].

Perhaps the most significant limitation is the inability to determine when the negative experiences being described in the tweets took place. The collected tweets were published online between 2015–2021 but could have taken place from minutes to decades before the tweet was published. There have been important changes to the approaches to delivering gender affirming treatment, who can deliver it (specialty 'gender' clinics versus primary care providers), and who is eligible to receive it. The provision of gender affirming care, and the language used to describe it, has evolved to be more person-centered and less pathologizing than it once was. For example, the diagnosis of "gender identity disorder" has become "gender dysphoria," shifting the focus of the problem from one's identity to the discomfort with one's body [98]. Similarly, the purpose of the current Standards of Care shifts the role of the healthcare professional from manager to assistant, includes transgender and gender diverse people, and considers an individual's overall health and wellbeing [99]. Similarly, in more recent years, a few states have made important strides in combating anti-transgender legislation by passing "shield" laws to assist out-of-state T+ folks, promoting access to GATs [100] or implementing policies to support T+ youth with unsupportive parents to consent to hormones [101]. Finally, President Biden's recent decision to reinstate the 2010 interpretation of "sex" in Section 1557 of the Affordable Care Act to include gender identity [102] went into effect in July 2024, making it illegal to discriminate against T+ people in healthcare settings [102].

However, structural and societal transphobia and assumptions about the binary nature of gender continue to permeate HC organizations and the individuals working in them. The debate of whether 'sex' is mutable [103] persists in science, law, and policy, and is evident in the numerous anti-LGBTQ bills being considered across the US [15]. Whether important gains such as the interpretation of "sex" in the ACA to include gender identity will remain are largely dependent on the results of the 2024 election [102]. Some of the restrictive legislation also has potential impacts on HCPs (e.g., facing charges, loss of license) that could create additional barriers to care. Even though some of the negative experiences reported in these tweets happened in an unknown time frame and there have been some significant policy changes in support of gender affirming care for T+ people, current literature suggests many T+ patients continue to experience discrimination and rejection in healthcare settings [104]. For example, in a 2022 US survey, of the nearly 66,500 transgender and nonbinary adults who reported receiving healthcare in the last 12 months, nearly 50% reported at least one negative experience in the healthcare setting [55]. As such, creating more inclusive healthcare environments is a priority that will require coordinated, multilevel efforts to improve factors ranging from

national, payor and institutional policies to HCP knowledge, attitudes and comfort working with T+ patients. The enforcement of inclusive policies and delivery of culturally sensitive care in direct clinical practice is critical [105]. The bulk of the recommendations that follow arise from the combination of our findings and existing research with similar findings and are most applicable to traditional healthcare settings; guidance has previously been provided on improving inpatient and residential mental health settings [106,107] and ancillary services such as the lab [108] and pharmacy [109].

**Implications for Clinical Practice.** At the organizational level, inclusive policies for T+ patients need to be developed, including rooming by gender rather than assigned sex in areas that are typically sex segregated [110]. When institutional processes (e.g., registration or transition from ED to inpatient unit) are at the root of such errors, HCPs must also advocate for appropriate changes at the institutional level to prevent continued harm to the T+ community. It is also necessary to ensure communication of important gender-related data is available between relevant departments in a healthcare system (e.g., pharmacy and lab). Provision and monitoring of gender affirming hormones should be more widely available in the primary care setting; a recent analysis of payor data indicated family practice was more frequently the prescribing HCP of GAT [111] and other work suggests individuals in family practice may be more open to providing care to T+ patients [112]. Provision of GAT using a truly informed consent model (absent of HCP's added conditions reinforcing restricted tropes of who is T+) [50] has the potential to increase T+ patients' autonomy and eliminating unnecessary steps to accessing life-saving, gender affirming care. HCPs can advocate for the inclusion of GATs in insurance coverage, correct misinformation about GATs, and speak out against the slate of legislation around the US that is seeking to limit or eliminate access to GAT, particularly among youth [113].

One recommendation to improve healthcare quality is to routinely collect sexual orientation and gender identity (SO/GI) across clinical settings, including specialty care settings [114,115]. However, as T+ individuals' tweets and existing literature suggest, creating physically and psychologically safe healthcare environments in which T+ individuals feel comfortable disclosing their gender identity and/or history is critical [116]. As such, these data should be collected in private clinical spaces rather than shared rooms divided by curtains or open desk areas (e.g., reception), [85] and by making it clear why the data are being collected and ensuring confidentiality [116]. When these data are truly needed for the healthcare interaction, explaining why and how the information will be use could decrease the perception that the questions are inappropriate and/or unnecessary. To help reduce the need for T+ patients to be asked for their name, pronouns, and other relevant information repeatedly, it would be helpful to create SO/GI data fields in the EHR that are more diverse in representation. One possible approach for this is having a free-text option and periodically reviewing the text being added in the description of "other" gender identities and creating drop-down entries based on those [40]. In addition to having a place to document this information, it should be easy to retrieve

**Implications for Healthcare Education.** There is increasing interest among some HCPs to provide care to T+ people, but a lack of training in how to do so [112]. Lack of content in healthcare training programs has been associated with higher levels of transphobia [117] and can influence various aspects of the care interaction, including intake and assessment processes, [85,118] language used, [33] and HCP-patient comfort [38]. Faculty who likely never received training on caring for T+ patients may not feel prepared to teach it themselves, [36] leaving health professions students un[der]prepared to provide appropriate care to LGBTQ+ patients [119]. Lack of representation in health training curriculum may also be a barrier to the retention of health professions' students who are members of the T+ community.

Cisheteronomative (at best) and transphobic (at worst) training environments also have a negative impact on the health professions students who are members of the T+ community [120]. Actively recruiting and supporting T+ health professions students is an important aspect of diversifying the healthcare workforce [121].

Some universities have begun to incorporate care of T+ patients in their curricula [122–126]. However, basic education may be still lacking [125] or knowledge improvements may not be sustained [126,127]. Further, education alone may not be enough. HCPs' transphobia is not only associated with lack of knowledge about care for T+ people, [128] but also less willingness to provide care for a T+ patient, [112] and may vary among HCP types [117]. Thus, it is imperative improve HCPs' attitudes about T+ people before providing new knowledge. Prior evidence suggests that opportunities for contact with T+ people through speakers' panels [129] or one-on-one interactions (e.g., simulation/standardized patients) [124,130] outside of the direct care environment can help improve comfort with T+ people. Similarly, inclusion of T+ individuals in the development of simulation scenarios and other learning activities would be helpful to represent the range of T+ patients' identities and lived experiences and may be more impactful for learners [131].

**Implications for Research.** Research about the health and healthcare needs of the T+ community must be conducted with meaningful engagement of T+ community members (e.g., community-based participatory approaches), in collaboration with T+ researchers to advance the health and healthcare experiences of T+ people, [132,133] and without reproducing traditional hierarchies [134]. Building on work identifying negative interactions and consequences, it could be helpful to explore the components of T+-affirming healthcare, expanding beyond treating with basic respect (e.g., using appropriate names and gender pronouns) to fully address the unique needs of this patient population.

## Conclusions

For nearly a decade, T+ people have used #TransHealthFail to describe negative healthcare experiences. The tweets highlighted antecedents such as perceptions of provider's lack of knowledge and rigid binary beliefs about gender and barriers ranging from having to educate the provider to get the care they needed to health insurance issues. The most common problems described were experiences of misgendering, invalidation, and exposure, which occurred across healthcare treatment settings and provider types. T+ people's tweets also indicated that they perceived structural barriers in healthcare systems, including what was experienced as administrative violence. T+ people described the impacts of negative interactions on care including that they felt their care needs were not adequately met or were incorrectly treated, which led to feelings of fear and decisions to avoid care. Finally, T+ individuals shared a range of strategies they used to help navigate healthcare systems and attempted to avoid negative interactions. A combination of these findings and existing research suggest a concerted, multi-pronged approach to facilitate changes is warranted to both reduce barriers to healthcare and improve T+ people's perceptions of their healthcare experiences. It is critical to include members of the T+ community with diverse identities and lived experiences in all aspects of solution development and implementation through collaboration with community organizations and T+ service providers and researchers.

## Acknowledgements

The authors would like to acknowledge the Institute for Health Equity and Social Justice at Northeastern University for the research training provided to AM, SN and LG as part of their Health Equity Intern program. JDB would like to thank the "Reckoning" Cohort of Humanities

Fellows at Northeastern University] and members of the Harvard Sexual Orientation, Gender Identity and Expression (SOGIE) Health Equity Research Collaborative for feedback on works-in-progress. The authors are also grateful for KJ Rawson and their team at the Digital Transgender Archivefor making access to materials about transgender history readily available.

## Author contributions

**Conceptualization:** Jordon D Bosse.

**Data curation:** Jordon D Bosse.

**Formal analysis:** Allison J. McLaughlin, Saren Nonoyama, Lauren Glupe, Jordon D Bosse.

**Funding acquisition:** Jordon D Bosse.

**Investigation:** Jordon D Bosse.

**Methodology:** Jordon D Bosse.

**Project administration:** Jordon D Bosse.

**Supervision:** Jordon D Bosse.

**Validation:** Jordon D Bosse.

**Visualization:** Saren Nonoyama, Jordon D Bosse.

**Writing – original draft:** Allison J. McLaughlin, Saren Nonoyama, Lauren Glupe, Jordon D Bosse.

**Writing – review & editing:** Allison J. McLaughlin, Saren Nonoyama, Lauren Glupe, Jordon D Bosse.

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
