## [Editor Report · Decision Letter 0]

1 Feb 2024

PDIG-D-23-00437

Systemic transphobia and ongoing barriers to healthcare for transgender and nonbinary clients: An analysis of #TransHealthFail

PLOS Digital Health

Dear Dr. Bosse,

Thank you for submitting your manuscript to PLOS Digital Health. After careful consideration, we feel that it has merit but does not fully meet PLOS Digital Health's publication criteria as it currently stands. Therefore, we invite you to submit a revised version of the manuscript that addresses the points raised during the review process.

Please submit your revised manuscript within 60 days Apr 01 2024 11:59PM. If you will need more time than this to complete your revisions, please reply to this message or contact the journal office at digitalhealth@plos.org. Please include the following items when submitting your revised manuscript:

We look forward to receiving your revised manuscript.

Kind regards,

Catherine Bielick, M.D., MSc, MSc

Guest Editor

PLOS Digital Health

Journal Requirements:

1. Since your data is not available for proprietary reasons, please explain via email why the data is not available. Please also include the contact information for the third party organization that should be contacted should other researchers want to request access to this data and please include the full citation of where the data can be found. We also request that you verify with us via email that any researcher will be able to obtain the data set in the same manner that the you have obtained it. If you feel you are unwilling or unable to adhere to this policy, please explain your reasons by return email and your exemption request will be escalated to the editor for approval. Your exemption request will be handled independently and will not hold up the peer review process, but will need to be resolved should your manuscript be accepted for publication. One of the Editorial team will be in touch if they require more information.

Additional Editor Comments (if provided):

This is a timely study which focuses on a topic at the intersection of social, ethical, and public health domains. There are very many results and conclusions here that have high value and pertinence to society as a whole. These strengths are worth exploring and do fit into the scope and mission of the journal, but there are also important matters that should be addressed before going to full review. 

The technique of utilizing a social media hashtag to pull a diverse and accessible range of experiences and stories seemed relatively novel and comprehensive in an unconventional way. One key limitation of this technique for such a dynamic and emotionally charged topic is that, while comments themselves were included for the years 2015-2021, there was not a description or limit described for when those experiences actually occurred. For instance, it seems like experiences could have been described from forty years ago when Real Life Experience was mandatory for securing gender affirming medical care and were aggregated with experiences from as recently as four years ago. The WPATH standards of care, society guidelines, the political landscape, and cultural acceptance have vastly changed in content and awareness during these decades. When did the experiences labelled as "Many reported that lack of health insurance coverage, which included blanket exclusions for GATs, often deeming them cosmetic or elective" occur in relation to Section 1557 of the Affordable Care Act? This limitation should be addressed or otherwise conclusions should be substantially tempered with this in mind. 

While there is no word count limit on manuscripts, the length related to diffusion of content is an issue. In the materials and methods/data analysis section, four categories are described: antecedents, barriers, outcomes, and consequences. But then in the results not only is a fifth category added, but one of those categories is further broadened into seven more subcategories with 3-5 described items within each of the twelve categories. Then an additional section of more categories continues to expand at the end. This will be overwhelming to reviewers and readers, and gives the sense that not enough distillation was performed. There also is currently no way of identifying which categories were more prevalent than others without an idea of the objective distribution of comments into each category. The reader must presume that each issue raised carried equal importance and prevalence to all others. With any restructuring, if the information is available then there may also be value in delineating a bit more some of the experiences and reports of non-binary transgender people from binary transgender people who may have different concerns from each other. Another option may include describing each group a bit more before going into the details of types of mistreatment. For instance, some binary trans men and women do not prefer to have pronouns frequently asked and may have had different "hoops" to jump through in demonstrating their new gender role. In contrast some non-binary transgender people may have frustratingly had to sell the "trapped in the wrong body and/or interest in a specific trajectory of medical transition" narrative which binary transgender people might be more comfortable with describing. 

Lastly there is a surplus of emotionally charged language including alleged legal culpability that should be re-examined before going to review. Examples include "the emergency medical technician stopped treating her when her transgender status became apparent and used derogatory language," "after being refused treatment by more than 20 doctors; by the time someone was willing to treat him, it was too late," "Places who should know better/places that should do better," "T+ care being seen as specialty care, requiring higher copayments," "Often HCP's justification for delaying care was arbitrary," "HCPs also sexualized T+ clients by lewd looks and making remarks about clients' attractiveness," and the use of the term "administrative violence." There is no doubt that the two cases above are great tragedies, but significant care should be made when commenting on legal conclusions related to criminal negligence by remaining as objective as possible. If including these cases is absolutely necessary to frame the study, then some ways around this may be to use terms that suggest possible culpability or reporting conclusions already made in a court of law rather than using language that commands it. For this and many reasons, some verbatim comments from the participants would be valuable to include where pertinent in the results. 

There are other notes which reviewers may raise that may be worth investigating before resubmission, but are not barriers to going to review. Without being able to read any of the statements objectively written, then it is hard to know if categories such as "comparing lab and test values to the wrong reference range" and "receiving the wrong medication" are related to patient perception or true medical error. For instance, a trans woman might believe that the creatinine range for non-transgender women should be used for herself in all cases and as such report this using the hashtag. But for a variety of physiologic reasons the UCSF Guidelines recommend using the upper limit of the non-transgender male value for creatinine, hemoglobin, and alkaline phosphatase. Were these potential discrepancies taken into account before coding this category? Other statements in a similar vein include "possibly contraindicated medications," "lower quality care," "not receiving accurate information or assessments," and "leaving ER visits without receiving adequate testing." By whose judgment were these conclusions made? The authors might take care not to communicate with language that overly commodifies healthcare as well. 

The study reports on a highly important topic and includes valuable results for the community at large. This feedback is not meant to be a discouragement, but hopefully can refine the manuscript in a way that it can reach the most people in the most effective way.
---

## [Decision Letter · Decision Letter 1]

9 Sep 2024

PDIG-D-23-00437R1

Systemic transphobia and ongoing barriers to healthcare for transgender and nonbinary clients: An analysis of #TransHealthFail

PLOS Digital Health

Dear Dr. Bosse,

Thank you for submitting your manuscript to PLOS Digital Health. After careful consideration, we feel that it has merit but does not fully meet PLOS Digital Health's publication criteria as it currently stands. Therefore, we invite you to submit a revised version of the manuscript that addresses the points raised during the review process.

Please submit your revised manuscript within 60 days Nov 08 2024 11:59PM. If you will need more time than this to complete your revisions, please reply to this message or contact the journal office at digitalhealth@plos.org. Please include the following items when submitting your revised manuscript:

We look forward to receiving your revised manuscript.

Kind regards,

Catherine G Bielick

Guest Editor

PLOS Digital Health

Journal Requirements:

Additional Editor Comments (if provided):

I'd like to apologize for the delay in processing peer reviews for this manuscript. The process of identifying peer reviewers who could be expected to approach this review with sensitivity and domain proficiency took time, and several turned out to be unavailable. 

The changes submitted in the last version were strong, and one reviewer was ready to accept. Another reviewer gave very valuable and substantive feedback which could strengthen the manuscript even further in a revision. I think the responses from this reviewer are reasonable in their request for edits to the language and editing of the discussion section. Some typographical errors remain, in addition to the ones in this reviewer's comment another is in line 351, "Our analysis sought categorized negative healthcare experiences..." Please take a look through as we are all in agreement that the subject matter and results are important and in line with the mission here.

Reviewers' comments:

Reviewer's Responses to Questions

**Comments to the Author**

1. If the authors have adequately addressed your comments raised in a previous round of review and you feel that this manuscript is now acceptable for publication, you may indicate that here to bypass the “Comments to the Author” section, enter your conflict of interest statement in the “Confidential to Editor” section, and submit your "Accept" recommendation.

Reviewer #1: (No Response)

Reviewer #2: All comments have been addressed

2. Does this manuscript meet PLOS Digital Health’s publication criteria ? Is the manuscript technically sound, and do the data support the conclusions? The manuscript must describe methodologically and ethically rigorous research with conclusions that are appropriately drawn based on the data presented.

Reviewer #1: Partly

Reviewer #2: Yes

3. Has the statistical analysis been performed appropriately and rigorously?

Reviewer #1: N/A

Reviewer #2: N/A

4. Have the authors made all data underlying the findings in their manuscript fully available (please refer to the Data Availability Statement at the start of the manuscript PDF file)?

Reviewer #1: No

Reviewer #2: Yes

5. Is the manuscript presented in an intelligible fashion and written in standard English?

PLOS Digital Health does not copyedit accepted manuscripts, so the language in submitted articles must be clear, correct, and unambiguous. Any typographical or grammatical errors should be corrected at revision, so please note any specific errors here.

Reviewer #1: Yes

Reviewer #2: Yes

6. Review Comments to the Author

Please use the space provided to explain your answers to the questions above. You may also include additional comments for the author, including concerns about dual publication, research ethics, or publication ethics. (Please upload your review as an attachment if it exceeds 20,000 characters)

Reviewer #1: PLOS Digital Health Review

Background: First sentence should probably say “transgender” in it somewhere, possibly the first word was lost?

p. 4 line 84 – Suggest change to “with some estimates SUGGESTING a higher proportion” the word ‘demonstrating’ following estimates seems inappropriate. 

p. 5 Should add time periods to the statements in this paragraph (i.e. when the data in the citations you’re using are from) in order to contextualize. Alternatively, you could cut the middle sentences (Lines 87-92) which would reduce timeliness concerns

Line 110 It feels like this needs another sentence to clarify if the authorship team is associated with My Trans Health and/or to discuss the longevity of the hashtag before starting with “The purpose of this project”

Lines 135-8 It would be clearer if you moved the number excluded after the description of why they were excluded. I.e. In all, 269 tweets were excluded, leaving 1340 …

Line 167 You have an extra “and” before Mississippi

Line 170 Is this supposed to be the most common needS WERE? Or are you saying mental health services inclusive of outpatient, inpatient and acute emergency? Needs clarifying

Line 174 Would say “T+people TWEETED ABOUT EXPERIENCING” rather than “T+ people experienced” While I agree with the conclusion, it’s not actually the data. There are additional comments about this below. Your data is about what was described not what happened. 

Line 177: The three most common HCP types WERE … OR you could say “the most common HCP types included….”

Line 178: Others NAMED AS responsible… 

Line 192 I believe “Little to know experience” is supposed to italicized as part of the theme? 

…. I’m stopping going line by line here, but the whole results section needs to be rewritten to be clear that you’re categorizing descriptions of experiences not the experiences themselves

Throughout these descriptions, it is not that the HCP’s lack of knowledge WAS an antecedent, it’s that tweets described HCPS as having a lack of knowledge, training, and experience, which was conceptualized as an antecedent. And that HCPs were experienced as holding rigid, binary gender beliefs. You don’t know the HCPs held those believes, the tweets described those beliefs as being held. 

Throughout the results there is also substantial discussion of the results that should be moved to the discussion which will allow you to put the results (themes) in the context of your knowledge of the fact that these types of experiences have been documented in different ways elsewhere. 

That said, the themes and quotes seem appropriate, but they need to be more accurately described as being about reports of experiences rather than the experiences. 

Discussion: 

Line 351 “ Our analysis SOUGHT TO CATEGORIZE T+ PEOPLE’S TWEETS ABOUT THEIR NEGATIVE…”

357: I think this point is very important that if providers present as competent, it’s worse when they’re not

387 highlight A PERCEPTION OF HCP’s AS HAVING outdated ideas… 

Again, in the discussion as in the results it is important to remember that you are describing tweets from T+ people describing their experiences of HCPs. You can not discuss those HCPs, only the perceptions of them. While I agree that those perceptions are likely accurate, the T+ patients’ experiences occurred based on the perception not necessarily the reality and would still occur even if the perception was not the reality. I’m not going line by line in the discussion, but it also needs a full rewrite keeping that in mind. Right now it reads more as a polemic than an accurate discussion of the research

This whole paper would be a lot stronger if the authors were clearer in their own minds about the fact that they are analyzing tweets not descriptions of individuals that are necessarily accurate. I think this is an incredibly important data set, but right now I do not feel like it is suitable for publication as it elides the difference between tweeters’ perception of experiences and the reality of the underlying factors. The expectation of negative behavior is still a problem and can be demonstrated as based on experiences of others from other studies, but tweets can not be assumed as accurate descriptors for any number of reasons including the fact that descriptions of healthcare issues may have nothing to do with the person being T+ but due to underlying problems with healthcare systems that affect many patients, but certainly more those with minoritized identities. 

Lines 410-421 If you’re going to start talking about diagnosis as a double edged sword you should also talk about the fact that the way that the US medical system works is through the treatment of diagnoses. If there is no gender diagnosis, how will patients be able to access treatment without a full rework of the medical system? I’m not saying that it’s right, but it is how it works – you need a diagnosis for care to be covered except for preventative health services. (Which is an interesting argument to attempt to make for gender affirming care but is likely beyond the scope of this piece)

The Implications sections seem out of scope for this work, except for the research paragraph. 

Conclusions are not actually about your study. Your conclusions are “T+ individuals describe a number of problematic interactions with the US healthcare system, which are supported by this historical literature. The most common problems described were: … There is a need to… 

The conceptual model diagram is great.

Reviewer #2: All my comments were answered very well. Inclusion of the Table 1 in its current form was a very substantive addition for qualitative research. The description of study limitations was thorough and respectful and while the discussion is still fairly long, it does a good job of reaching far into the implications of this work.

7. PLOS authors have the option to publish the peer review history of their article (what does this mean? ). If published, this will include your full peer review and any attached files.

**Do you want your identity to be public for this peer review?** For information about this choice, including consent withdrawal, please see our Privacy Policy . 

Reviewer #1: No

Reviewer #2: Yes: Catherine Grace Bielick

---

## [Editor Report · Decision Letter 2]

9 Dec 2024

Systemic transphobia and ongoing barriers to healthcare for transgender and nonbinary clients: An analysis of #TransHealthFail

PDIG-D-23-00437R2

Dear Dr. Bosse,

We are pleased to inform you that your manuscript 'Systemic transphobia and ongoing barriers to healthcare for transgender and nonbinary clients: An analysis of #TransHealthFail' has been provisionally accepted for publication in PLOS Digital Health.

Best regards,

Laura Sbaffi, PhD, MA, MSc

Section Editor

PLOS Digital Health